# Enhancing the Therapeutic Effect of 2-^211^At-astato-α-methyl-L-phenylalanine with Probenecid Loading

**DOI:** 10.3390/cancers13215514

**Published:** 2021-11-03

**Authors:** Hirofumi Hanaoka, Yasuhiro Ohshima, Hiroyuki Suzuki, Ichiro Sasaki, Tadashi Watabe, Kazuhiro Ooe, Shigeki Watanabe, Noriko S. Ishioka

**Affiliations:** 1Faculty of Medicine, Kansai Medical University, 2-5-1 Shin-machi, Hirakata 573-1010, Osaka, Japan; 2Department of Radiotheranostics, Gunma University Graduate School of Medicine, 3-39-22 Showa-machi, Maebashi 371-8511, Gunma, Japan; 3Department of Radiation-Applied Biology Research, Quantum Beam Science Research Directorate, National Institute for Quantum Science and Technology, 1233 Watanuki-machi, Takasaki 370-1292, Gunma, Japan; ohshima.yasuhiro@qst.go.jp (Y.O.); sasaki.ichiro@qst.go.jp (I.S.); watanabe.shigeki@qst.go.jp (S.W.); ishioka.noriko@qst.go.jp (N.S.I.); 4Department of Molecular Imaging and Radiotherapy, Graduate School of Pharmaceutical Science, Chiba University, 1-8-1 Inohana, Chuo-ku, Chiba 260-8675, Chiba, Japan; h.suzuki@chiba-u.jp; 5Department of Nuclear Medicine and Tracer Kinetics, Graduate School of Medicine, Osaka University, 1-1 Yamadaoka, Suita 565-0871, Osaka, Japan; watabe@tracer.med.osaka-u.ac.jp (T.W.); ooe@tracer.med.osaka-u.ac.jp (K.O.)

**Keywords:** 2-^211^At-astato-α-methyl-L-phenylalanine, targeted alpha therapy, tumor retention, probenecid, blood clearance

## Abstract

**Simple Summary:**

To enhance the therapeutic effect of 2-^211^At-astato-α-methyl-L-phenylalanine (2-^211^At-AAMP), a radiopharmaceutical for targeted alpha therapy, we evaluated the effect of probenecid loading on its biodistribution and therapeutic effect in mice. Probenecid preloading significantly delayed the clearance of 2-^211^At-AAMP from the blood, increasing its accumulation in tumors. Consequently, the therapeutic effect of 2-^211^At-AAMP markedly improved. These results indicate that 2-^211^At-AAMP with probenecid loading is useful for the treatment of various types of cancers.

**Abstract:**

L-type amino acid transporter 1 (LAT1) might be a useful target for tumor therapy since it is highly expressed in various types of cancers. We previously developed an astatine-211 (^211^At)-labeled amino acid derivative, 2-^211^At-astato-α-methyl-L-phenylalanine (2-^211^At-AAMP), and demonstrated its therapeutic potential for LAT1-positive cancers. However, the therapeutic effect of 2-^211^At-AAMP was insufficient, probably due to its low tumor retention. The preloading of probenecid, an organic anion transporter inhibitor, can delay the clearance of some amino acid tracers from the blood and consequently increase their accumulation in tumors. In this study, we evaluated the effect of probenecid preloading on the biodistribution and therapeutic effect of 2-^211^At-AAMP in mice. In biodistribution studies, the blood radioactivity of 2-^211^At-AAMP significantly increased with probenecid preloading. Consequently, the accumulation of 2-^211^At-AAMP in tumors was significantly higher with probenecid than without probenecid loading. In a therapeutic study, tumor growth was suppressed by 2-^211^At-AAMP with probenecid, and the tumor volume was significantly lower in the treatment group than in the untreated control group from day 2 to day 30 (end of the follow-up period) after treatment. These results indicate that probenecid loading could improve the therapeutic effect of 2-^211^At-AAMP by increasing its accumulation in tumors.

## 1. Introduction

Targeted alpha therapy (TAT), a cancer treatment with a specifically delivered α-emitter, is an attractive potential therapy because of its high therapeutic effect without significant toxicity [1,2,3]. Many clinical studies of TAT targeting cancer-specific molecules have shown promising results [3,4]. However, the application of TAT is limited to certain types of cancers since most of the current target molecules are expressed in only some types of cancers. To widen the benefit of TAT, it is essential to select target molecules that are expressed in various types of cancers.

To sustain abnormal proliferation, several types of amino acid transporters are overexpressed in cancer cells [5,6]. Among them, L-type amino acid transporter 1 (LAT1) plays a significant role in cancer growth; hence, LAT1 is highly expressed in various types of human cancers [7,8]. Additionally, a LAT1-specific amino acid tracer, fluorine-18 (^18^F)-labeled α-methyl-L-tyrosine (^18^F-FAMT, Figure 1a), has shown specific accumulation in many types of malignant tumors, such as brain tumors, lung cancer, maxillofacial cancer, and thoracic cancer, in clinical practice [9,10]. Therefore, LAT1 is an ideal target of TAT for a broad range of cancers.

We previously developed α-methyl-L-phenylalanine derivatives labeled with ^18^F or bromine-76 (2-^18^F-FAMP and 2-^76^Br-BAMP, respectively; Figure 1b) as promising amino acid tracers targeting LAT1 [11,12]. Thereafter, we developed α-methyl-L-phenylalanine derivatives labeled with astatine-211 (^211^At; half-life, 7.2 h), an attractive radiohalogen α-emitter of halogen (2-^211^At-astato-α-methyl-L-phenylalanine [2-^211^At-AAMP]; Figure 1b) [13]. 2-^211^At-AAMP showed LAT1-specific cellular uptake and inhibited cell growth in vitro. It also accumulated in tumors and was shown to have a beneficial effect on survival in tumor-bearing mice [13]. However, its therapeutic effect was insufficient, likely due to its low tumor retention. Its therapeutic effect could be improved by increasing its accumulation and retention in tumors.

Since the tumor uptake of an amino acid targeting LAT1 is associated with its blood concentration [11,14], maintaining the blood radioactivity level could be an effective strategy to increase the accumulation of 2-^211^At-AAMP in tumors. Since 2-^211^At-AAMP is excreted in urine in an intact form [13], preventing its renal uptake could increase and maintain its accumulation level in tumors by delaying clearance from the blood. A previous study demonstrated that preloading of probenecid, an organic anion transporter (OAT) inhibitor, markedly delayed the clearance of radioiodine-labeled α-methyltyrosine (IMT; Figure 1a) from the blood and increased its accumulation in tumors by reducing renal uptake [15]. Considering the common target molecule (LAT1) and structural similarity between 2-^211^At-AAMP and IMT (Figure 1), we speculated that the probenecid loading strategy should work on 2-^211^At-AAMP. In this study, we evaluated the effect of probenecid loading on the biodistribution and therapeutic effect of 2-^211^At-AAMP in mice.

## 2. Materials and Methods

### 2.1. General

^211^At was produced via the ^209^Bi(α,2n)^211^At reaction and isolated by dry distillation [16]. Then 2-^211^At-AAMP was prepared from the stannyl precursor as previously described [13]. After purification via reversed-phase high-performance liquid chromatography, 2-^211^At-AAMP was obtained with high radiochemical purity (>95%). Probenecid was purchased from Tokyo Chemical Industry (Tokyo, Japan) and dissolved in 1 M NaOH solution. The pH of the solution was adjusted to approximately 8 using 1 M HCl and 1 M phosphate buffer (pH 7.4) and finally diluted with water in order to become an isotonic solution ready for injection. A human ovarian adenocarcinoma cell line, SKOV3, a human lung carcinoma cell line, A549, a human colon adenocarcinoma cell line, LS180, and a human glioblastoma cell line, U87MG, were purchased from American Type Culture Collection (ATCC, Manassas, VA, USA). Six-week-old male ICR mice were purchased from Japan SLC (Hamamatsu, Japan). Five-week-old female BALB/c nude mice were purchased from CLEA Japan (Tokyo, Japan).

### 2.2. Biodistribution Studies

The animal experiments were approved by the Institutional Animal Care and Use Committee of the National Institute for Quantum Science and Technology (19-T001-1 and 20-T001-1), and all animal experiments were conducted in accordance with the institutional guidelines regarding animal care and handling.

For the biodistribution study in normal mice (6-week-old male ICR mice), probenecid solution was injected intraperitoneally (400 mg/kg/300 µL) 1 h before intravenous injection of 2-^211^At-AAMP (100 kBq) into the mice (*n* = 4 per group). The mice were euthanized, and the tissues of interest were excised and weighed at 10 min, 1 h, 3 h, and 6 h after the injection of 2-^211^At-AAMP. Saline was intraperitoneally preinjected in the control mice. The radioactivity was measured using a well-type gamma counter (ARC7001; Hitachi-Aloka Medical, Tokyo or 2480 Wizard2; PerkinElmer, Waltham, MA, USA). The uptake of the tracer was expressed as a percentage of the injected dose per organ (for the thyroid and stomach) or per gram of the organ (for other organs). Radiation-effective doses for humans were calculated from the biodistribution data of normal mice using OLINDA/EXM software (version 1.1; Vanderbilt University, Nashville, TN, USA) [17].

For the biodistribution study in tumor-bearing mice, SKOV3 cells (5 × 10^6^ cells/head) were inoculated into the right thigh of female BALB/c nude mice. When palpable tumors developed, 2-^211^At-AAMP (100 kBq) in 100 μL of phosphate buffered saline (PBS) was injected into the tail veins of the mice (*n* = 4 to 5 per group) 1 h after intraperitoneal preloading of probenecid solution (400 mg/kg/200 µL). The mice were euthanized 1 h, 3 h, and 6 h after the administration of 2-^211^At-AAMP, and the tissues of interest were excised and weighed. The radioactivity was measured using a well-type gamma counter as described above. The uptake of the tracer was expressed as a percentage of the injected dose per gram of the organ.

### 2.3. LAT1 Expression Analysis

To determine the effect of probenecid on LAT1 expression levels, in vitro Western blotting studies were performed. After treatment with 1 mM probenecid for 1, 3, or 6 h, cancer cells were dissolved in sample buffer containing 100 mM dithiothreitol and incubated at 65 °C for 15 min. Aliquots of samples were analyzed using SDS-polyacrylamide gel electrophoresis (Any kD Mini-PROTEAN TGX Precast Protein Gels, BioRad, Hercules, CA, USA) and transferred onto a polyvinylidene difluoride membrane. The blots were incubated at 4 °C overnight in 10 mM Tris–HCl, 100 mM NaCl, and 0.1% Tween 20 (pH 7.5; TBST), with 1% bovine serum albumin. Then the blots were incubated with mouse anti-LAT1 antibody (sc-374232, Santa Cruz Biotechnology, Dallas, TX, USA) or rabbit anti-β-actin antibody (#4970, Cell Signaling Technology, Danvers, MA, USA) at 4 °C overnight. After washing with TBST, the blots were incubated with horseradish peroxidase conjugated anti-rabbit or anti-mouse IgG antibodies at room temperature for 1.5 h. The blots were further washed with TBST, and specific proteins were visualized using Clarity Western ECL Substrate (BioRad). The expression levels of LAT1 in the xenograft tumors were also analyzed using Western blotting. SKOV3 tumors excised from eight tumor-bearing mice were homogenized using BioMasher II (Nippi, Tokyo, Japan) in a lysis buffer. After centrifugation at 15,000× *g* for 30 min, the supernatant was mixed in a sample buffer containing 100 mM dithiothreitol and incubated at 65 °C for 15 min. Aliquots of samples were analyzed as described above. The original Western Blots data was shown in the Appendix A.

### 2.4. Therapeutic Study

Tumor-bearing mice were prepared in the same manner as that in the biodistribution study. When the tumors were fully established (141.8 ± 46.9 mm^3^ in size), a therapeutic study was conducted. 2-^211^At-AAMP (2 MBq/head) or PBS was intravenously injected into the mice 1 h after intraperitoneal preloading of probenecid solution (400 mg/kg/200 µL) at day 0. PBS injection without probenecid preloading was used in the control group. There was no significant difference in the initial tumor volume among the groups (*n* = 5 to 6 per group). The body weight and tumor volume were measured at least twice a week for 30 days. The tumor volume (mm^3^) was calculated from caliper measurements as (length × width^2^)/2, and the tumor volume was expressed relative to the initial tumor volume. In the case of weight loss >20%, moribund state signs, or a tumor volume >800 mm^3^, the mouse was euthanized humanely using isoflurane inhalation.

### 2.5. Statistical Analysis

Statistical analyses were performed using GraphPad Prism 9 (Graph Pad Software, San Diego, CA, USA). The results were expressed as the mean ± standard deviation (SD). Significant differences between two groups were determined using the unpaired *t*-test, and differences among more than two groups were determined using one-way analysis of variance (ANOVA) followed by Dunnett’s test. The survival curve was estimated in each group using Kaplan–Meier survival analysis, and the results were compared using the log-rank test. Differences were considered statistically significant when the *p*-value was <0.05.

## 3. Results

### 3.1. Biodistribution Study

To evaluate the effect of probenecid loading, we first conducted a biodistribution study in normal mice. As shown in Figure 2, the renal accumulation of 2-^211^At-AAMP was significantly lower at 10 min and higher at 1 h and 3 h after injection in the probenecid preloading group compared with that in the control group (without probenecid preloading). This is probably because probenecid inhibited the renal accumulation of 2-^211^At-AAMP at an early point in time, which delayed renal excretion. Consequently, the blood radioactivity of 2-^211^At-AAMP in the probenecid loading group was significantly increased compared with that in the control group at 1 h and 3 h after injection (*p* < 0.01). In mice, the pancreas expresses LAT1 [18] and can thus be regarded as an indicator for the tumor uptake of 2-^211^At-AAMP. Probenecid preloading significantly increased the accumulation of 2-^211^At-AAMP in the pancreas at 1 h and 3 h after injection (*p* < 0.01). Probenecid preloading also increased the accumulation of 2-^211^At-AAMP in other organs at 1 h and 3 h after injection owing to delayed clearance from the blood. In contrast, no significant difference was observed among all organs at 6 h after injection. The thyroid accumulation level in the probenecid preloading group was significantly higher than that in the control group at 3 h after injection (*p* < 0.05; Figure 3). However, since the difference disappeared at 6 h after injection, it was caused by delayed clearance from the blood rather than the specific accumulation of free ^211^At, which is highly retained in the thyroid [19].

The effective doses of 2-^211^At-AAMP with and without probenecid estimated from biodistribution data were 0.488 and 0.258 mSv/MBq, respectively. The estimated absorbed doses of 2-^211^At-AAMP with and without probenecid in the kidney were 0.079 and 0.048 mSv/MBq, respectively.

Figure 4 shows the biodistribution of 2-^211^At-AAMP in SKOV3-bearing mice. Probenecid preloading delayed the clearance of 2-^211^At-AAMP from the blood, similar to that in normal mice. The accumulation of 2-^211^At-AAMP in tumors in the probenecid loading group was significantly higher than that in the control group at 1 h and 3 h after injection (*p* < 0.01 and *p* < 0.05, respectively). Thereafter, in the probenecid loading group, radioactivity was rapidly eliminated from the tumors, and the accumulation level was similar to that in the control group at 6 h after injection. Probenecid preloading also increased the accumulation of 2-^211^At-AAMP in other organs at 1 h and 3 h after injection and then the accumulation became very low at 6 h after injection.

### 3.2. LAT1 Expression Analysis

The LAT1 expression levels were different in the four tumor cell lines analyzed (Figure 5a). Probenecid treatment for 6 h did not alter the expression level of LAT1 regardless of the cell line. LAT1 expression was also observed in the SKOV3 tumors (Figure 5b). The differences in expression levels among the mice could be due to the individual differences of the mice. In addition, tumor heterogeneity might have affected the results since a portion of the tumor was used instead of the whole tumor.

### 3.3. Therapeutic Effect in Tumor-Bearing Mice

To compare the therapeutic effect of 2-^211^At-AAMP with probenecid preloading with that of 2-^211^At-AAMP without probenecid preloading conducted in a previous study, we performed a therapeutic study with 2 MBq of 2-^211^At-AAMP [13]. Tumor growth was suppressed by 2-^211^At-AAMP treatment, whereas tumors treated with probenecid alone rapidly grew at the same rate as those in the control mice (Figure 6a). The tumor volume in all 2-^211^At-AAMP-injected mice (5/5 mice) temporarily decreased after treatment, and four of the five mice maintained tumor growth-free survival for >2 weeks (Figure 6b). The tumor volume was significantly lower in the 2-^211^At-AAMP-injected group than in the control group from day 2 to day 30 (end of the follow-up period) after treatment. The body weight of the mice transiently reduced (by <10%) after treatment with 2-^211^At-AAMP (Figure 6c), but it returned to normal levels within 2 weeks after treatment. In addition, since the change in body weight was similar between the 2-^211^At-AAMP and probenecid-only groups, the loss in body weight was caused by probenecid injection and not 2-^211^At-AAMP. Kaplan–Meier survival analysis revealed that the survival of mice significantly improved with 2-^211^At-AAMP treatment (*p* < 0.05; Figure 6d). No tumor volume became >800 mm^3^ (the euthanasia standards) in the 2-^211^At-AAMP treatment group for 30 days.

## 4. Discussion

We previously developed 2-^211^At-AAMP and demonstrated its potential as a radiopharmaceutical for TAT. However, the therapeutic effect of 2-^211^At-AAMP was found to be insufficient, probably due to its low tumor retention. Other ^211^At-labeled amino acids, including ^211^At-labeled L-phenylalanine (^211^At-Phe; Figure 1c) and ^211^At-labeled α-methyl-L-tyrosine (^211^At-AAMT; Figure 1a), have been successfully developed and have suppressed tumor growth in vivo [20,21,22,23]. However, there is concern about toxicity by irradiation following treatment with ^211^At-Phe or ^211^At-AAMT since they have shown a certain level of accumulation in normal organs. ^211^At-Phe was found to be taken up via not only LAT1 but also LAT2, which is expressed in normal organs [22]. ^211^At-AAMT is a LAT1-specific radiopharmaceutical, but it showed high renal accumulation in the excretion process [23]. In addition, dehalogenation to release free ^211^At was observed after the injection of ^211^At-Phe and ^211^At-AAMT, which can cause radiation exposure in the thyroid and stomach. Considering radiotoxicity, 2-^211^At-AAMP is an attractive radiopharmaceutical for TAT since it exhibits LAT1 specificity, has high in vivo stability, and shows low accumulation in normal organs.

The biodistribution study revealed that probenecid preloading prevented the renal uptake of 2-^211^At-AAMP, which delayed its clearance from the blood as expected. Consequently, the accumulation of 2-^211^At-AAMP in the pancreas, a LAT1-positive organ, or tumor, was significantly increased. Thus, probenecid preloading is a promising strategy to increase the accumulation of 2-^211^At-AAMP in tumors. The dose of probenecid (400 mg/kg) administered in this study is high compared with the normal human dose. However, we believe that probenecid administered at normal doses has a similar effect on 2-^211^At-AAMP in clinical practice since it is well known that a normal dose of probenecid (2000 mg/day) can prolong the plasma half-life and increase serum concentration of penicillin derivatives in humans. On the other hand, the effect of probenecid was lost 6 h after injection, which could be attributed to the elimination of probenecid from the body after a certain period. However, probenecid is generally repeatedly administered perorally in clinical practice, which could sustain its effect. Indeed, it has been reported that repeated peroral administration of probenecid at a normal dose for several days can maintain its plasma concentration and result in a significant increase in the plasma concentration of the other co-administered medication in patients [24]. Thus, repeated peroral administration of probenecid could further delay clearance from the blood and continuously increase the accumulation of 2-^211^At-AAMP in tumors. We selected the intraperitoneal route of probenecid administration according to our results with ^18^F-FAMT, but we confirmed that the effect of peroral probenecid administration was similar to that of intraperitoneal probenecid administration in that study (submitted for review).

Probenecid is an OAT inhibitor, but there is a possibility that it affects LAT1 expression levels in tumor cells or the uptake of 2-^211^At-AAMP into tumor cells. Our Western blotting results revealed that probenecid treatment for 6 h did not alter the LAT1 expression levels in SKOV3 cells. In addition, it has been reported that cellular uptake of IMT did not change with the addition of probenecid [15]. Thus, the increase in tumor accumulation levels with probenecid preloading is not caused by a direct effect on tumor cells but reflects delayed blood clearance. Furthermore, probenecid treatment did not change the LAT1 expression levels in other cell lines. These results indicate that the method involving 2-^211^At-AAMP with probenecid loading could be applied to other LAT1-positive cancers.

An increase in blood radioactivity would increase radiation exposure to the body. The effective dose for humans calculated from mice was increased with probenecid preloading. However, the toxicity of TAT has been reported to be low, even when radiolabeled antibodies with very slow clearance from the blood are administered [25]. Renal toxicity is another concern of TAT since many small-sized radiopharmaceuticals exhibit high renal accumulation and retention [26,27]. However, this is yet to be explored since the renal accumulation of 2-^211^At-AAMP is originally low and probenecid loading does not increase renal retention but inhibits renal uptake of 2-^211^At-AAMP. Indeed, the estimated absorbed dose in the kidney was increased only 1.65 times by probenecid preloading. Since probenecid loading can reduce the radioactivity dose of 2-^211^At-AAMP, probenecid loading is unlikely to increase renal toxicity. Pancreatic toxicity is another concern since 2-^211^At-AAMP showed high pancreatic accumulation in mice. However, high pancreatic accumulation is not expected in patients since the human pancreas lacks LAT1 expression [28]. Indeed, ^18^F-FAMT was shown to have low accumulation in the human pancreas in a clinical setting [9] despite high accumulation in the murine pancreas [29]. A delay in the clearance of 2-^211^At-AAMP from the body would increase the risk of dehalogenation to release free ^211^At, which could cause side effects [19,30]. In the biodistribution study, the accumulation level in the stomach or thyroid was very low at all measured time points, indicating that 2-^211^At-AAMP is stable for at least several hours. Since no increasing trend of accumulation in the stomach or thyroid was observed by probenecid loading, further delay in the clearance of 2-^211^At-AAMP from the blood would not cause dehalogenation. Based on these facts, probenecid loading does not increase the risk of toxicity.

In the therapeutic study, a sufficient therapeutic effect was observed following injection of 2-^211^At-AAMP with probenecid preloading. Importantly, the tumor volume in all five mice temporarily decreased after treatment and four of five mice maintained tumor growth-free survival for >2 weeks. In contrast, only two of five mice showed stable disease in a previous study involving treatment with 2-^211^At-AAMP without probenecid loading [13]. Since the therapeutic effect of TAT should depend on the radiation dose to the tumor, the improvement in the therapeutic effect is caused by increasing accumulation and retention in the tumor through probenecid preloading. Thus, probenecid loading is an encouraging strategy to improve the therapeutic effect of 2-^211^At-AAMP. There was no apparent increase in the side effects following probenecid preloading. Although we did not evaluate the toxicity in detail, the current dose would not cause severe toxicity since no weight loss was observed. In a previous study, meta-^211^At-astatobenzylguanidine (^211^At-MABG) caused temporary weight loss but no marked damage to the bone marrow with injection of the maximum tolerated dose in nude mice [31]. In a normal mice study, transient body weight loss, leucopenia, and renal damage were observed on injection of ^211^At-MABG; however, these toxicities were not severe, and the mice recovered by day 28 [32]. In addition, since 2-^211^At-AAMP shows rapid clearance from the body, radioactivity in the body was low at 6 h after injection, even in the probenecid-loading group. The accumulation levels in the blood and kidney were <0.2% dose/g and <0.5% dose/g at 6 h after injection, respectively, which were much lower than that for free ^211^At or ^211^At-MABG [19,31,32]. Since the half-life of ^211^At is 7.2 h, less retention of radiopharmaceuticals in non-target organs is important to reduce radiotoxicity. Indeed, the estimated absorbed dose of 2-^211^At-AAMP in the kidney is much lower than the reported value for free ^211^At or ^211^At-MABG [19]. Probenecid causes temporary weight loss. However, since probenecid will be used at a normal dose in clinical practice, it is not expected to cause severe toxicity. Although the evaluation of radiotoxicity in detail is necessary, the toxicity of 2-^211^At-AAMP with probenecid loading is expected to be low in view of these facts.

The administration of a higher radioactivity dose of 2-^211^At-AAMP would also improve the therapeutic effect. However, a higher dose would increase the renal radiation dose, which may cause radiotoxicity. Moreover, since the available radioactivity of ^211^At is limited and the radiochemical yield of 2-^211^At-AAMP is low (approximately 20%) [13], probenecid loading is a better strategy than using a higher dose when considering the number of patients treated with 2-^211^At-AAMP. Furthermore, with probenecid loading, it is possible to control the pharmacokinetics of 2-^211^At-AAMP and maintain an ideal situation by optimizing the probenecid loading plan (dose and schedule). After maintaining the blood radioactivity level of 2-^211^At-AAMP with probenecid loading for a certain period of time according to the half-life of ^211^At, radioactivity could be quickly cleared from the body by reducing the blood level of probenecid, which would provide maximum therapeutic effect with minimum toxicity. Since we have already developed radiolabeled α-methyl-L-phenylalanine as an imaging agent, the appropriate plan of probenecid loading for each patient could be determined by using this tracer.

There were some limitations to this study. First, we used only one cell line for the biodistribution and therapeutic studies. Since the availability of ^211^At is limited, it was difficult to perform a therapeutic study with other cell lines. Although the use of other cell lines might have influenced the therapeutic effect, it would have been irrelevant to the delayed blood clearance effect and the tumor accumulation enhancement effect by probenecid loading. Therefore, the probenecid preloading strategy has great potential to enhance the therapeutic effect of 2-^211^At-AAMP for any type of LAT1-positive tumor. Second, we did not directly compare the therapeutic effect with or without probenecid loading. Because of the limited available radioactivity of ^211^At and the low radiochemical yield of 2-^211^At-AAMP, we could only prepare approximately 10 MBq of 2-^211^At-AAMP in the current situation, which could cover only one 2-^211^At-AAMP treated group (injecting 2 MBq/head). As the accumulation of 2-^211^At-AAMP in tumors significantly increased with probenecid loading and 2-^211^At-AAMP showed a marked therapeutic effect with probenecid loading, the usefulness of the probenecid preloading strategy is unquestionable. Third, we followed the mice for only 30 days after treatment, based on the permission received from the facility. However, 30 days is sufficient to show the therapeutic effect of 2-^211^At-AAMP with probenecid loading. Considering the results presented in Figure 6, a longer observation period would provide better data to support the findings.

## 5. Conclusions

Probenecid preloading significantly delayed the clearance of 2-^211^At-AAMP from the blood and consequently increased its accumulation in tumors. Probenecid preloading could improve the therapeutic effect of 2-^211^At-AAMP by increasing its accumulation in tumors. These results indicate that 2-^211^At-AAMP with probenecid preloading is useful for the treatment of LAT1-positive cancers.

## Figures and Tables

**Figure 1 cancers-13-05514-f001:**
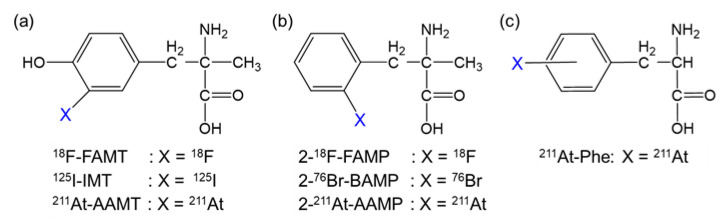
Structure of radiohalogen-labeled amino acids targeting LAT1. Radiohalogen-labeled (**a**) α-methyl-L-tyrosine, (**b**) α-methyl-L-phenylalanine, and (**c**) L-phenylalanine derivatives.

**Figure 2 cancers-13-05514-f002:**
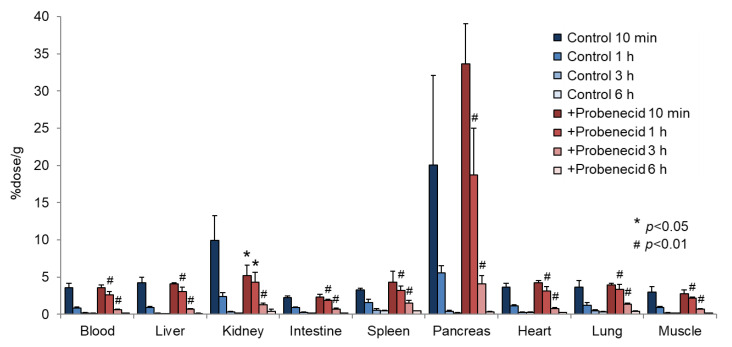
Biodistribution of 2-^211^At-AAMP in normal mice with or without intraperitoneal injection of probenecid (mean ± SD, *n* = 4). * *p* < 0.05 and ^#^ *p* < 0.01 compared with the control group (without probenecid preloading).

**Figure 3 cancers-13-05514-f003:**
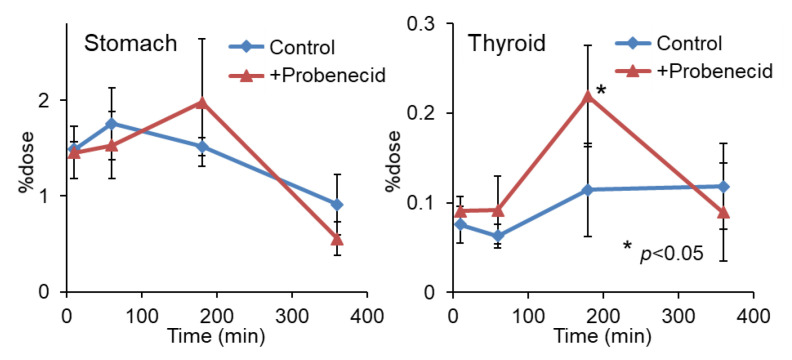
Stomach and thyroid radioactivity levels after injection of 2-^211^At-AAMP in normal mice with or without intraperitoneal injection of probenecid (mean ± SD, *n* = 4). * *p* < 0.05 compared with the control group (without probenecid preloading).

**Figure 4 cancers-13-05514-f004:**
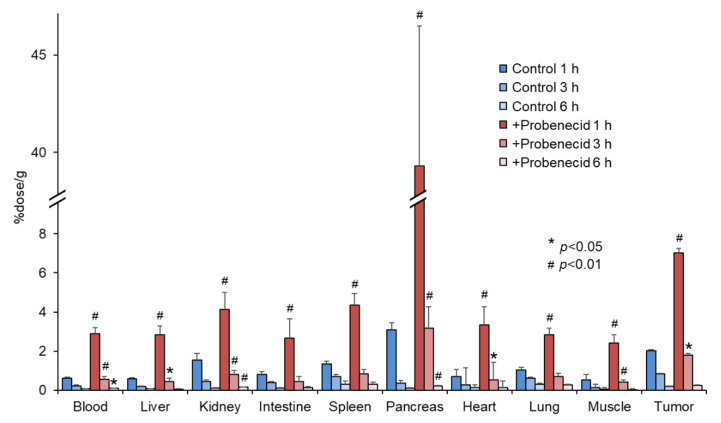
Biodistribution of 2-^211^At-AAMP in tumor-bearing mice with or without intraperitoneal injection of probenecid (mean ± SD, *n* = 4–5). * *p* < 0.05 and ^#^ *p* < 0.01 compared with the control group (without probenecid preloading).

**Figure 5 cancers-13-05514-f005:**
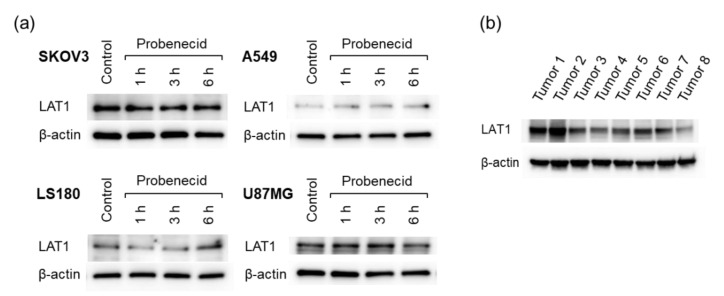
LAT1 expression in cell lines and excised tumors. (**a**) Four cell lines (SKOV3, A549, LS180, and U87MG) treated with probenecid (1 mM) and (**b**) SKOV3 tumors excised from eight tumor-bearing mice.

**Figure 6 cancers-13-05514-f006:**
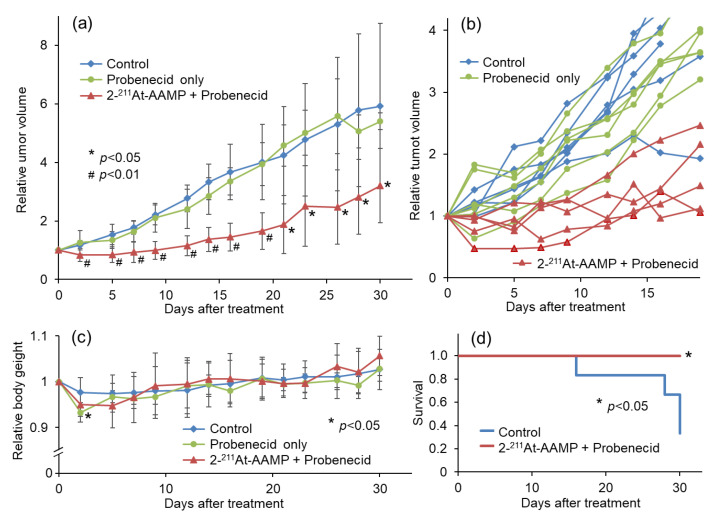
Therapeutic effect of 2 MBq 2-^211^At-AAMP in tumor-bearing mice with or without intraperitoneal injection of probenecid (mean ± SD, *n* = 5–6). (**a**) Average tumor growth, (**b**) individual tumor growth, (**c**) average body weight change, and (**d**) Kaplan–Meier survival curves after treatment. * *p* < 0.05 and ^#^ *p* < 0.01 compared with the control group.

## Data Availability

The data presented in this study are available in the article.

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
