# Peer review of "Enhancing the Therapeutic Effect of 2-211At-astato-α-methyl-L-phenylalanine with Probenecid Loading"

_cancers, 2021, doi:10.3390/cancers13215514_

Round 1
Reviewer 1 Report
authors did have read very well my suggestions, remarks, grammatical typos and did the necessary corrections,
further questions - f.i regarding LAT presence in pancreas in mice and concern for human therapy - was good motivated.
I do not have any remarks anymore of the revised paper,
so, for my feelings this manuscript can be published.
Reviewer 2 Report
The authors successfully replied to my major concerns.
This manuscript is a resubmission of an earlier submission. The following is a list of the peer review reports and author responses from that submission.
Round 1
Reviewer 1 Report
This is a refreshingly well-written manuscript based on a well-designed experimental plan. The authors appropriately interpret the results and identify potential deficiencies in the work that could be further examined in future studies. The results are significant because they identify a potential strategy for increasing tumor dose of the proposed agents and identify also potential toxicities that might arise. The manuscript will be of interest to a broad spectrum of investigators pursuing similar TAT research. The manuscript should be reviewed for minor grammatical errors, but otherwise is ready for publication.
Author Response
This is a refreshingly well-written manuscript based on a well-designed experimental plan. The authors appropriately interpret the results and identify potential deficiencies in the work that could be further examined in future studies. The results are significant because they identify a potential strategy for increasing tumor dose of the proposed agents and identify also potential toxicities that might arise. The manuscript will be of interest to a broad spectrum of investigators pursuing similar TAT research. The manuscript should be reviewed for minor grammatical errors, but otherwise is ready for publication.
Reply:
Thank you for the high rating. We apologize for the grammatical errors. The manuscript has been proofread by an English expert. We are also attaching the certificate.

Reviewer 2 Report
major;
(i) line 152-153 : statement: 'the pancreas expresses LAT1', is this really so ?, can this be underpinned by some references , [some references state, increased uptake in pancreas, for F18 labelled AA (Laverman et al., Eur J Nucl Med 2002, 29, 681 and Hanoaka, J Nucl Med...) but not always found ,everywhere ! ( i.e. *I-IPP, Baum et al., Nucl Med Mol Imaging 2011, (45), 299-307) ] - this should also mean, using *At-AAMP for cancer therapy, the pancreas will be exposed to a high radiation burden ! (or maybe useful for treatment of pancreatic adenocarcinoma, as there are some references in literature, with radio-labelled modified amino acids),
(ii) can probenecid really be used for clinical studies ?, can the translation be really made ? (as dose of probenecid, for mice is now 400 mg/kg), what is the opinion of the authors ?
minor :
line 22 : is useful as adjuvant...,
line 43 : delete 'anti-cancer' ,
line 45 : promising (instead of great),
line 63 : ... an attractive radiohalogen α-emitter...,
line 76 : the clearance of radioiodine-labeled...,
line 90-91 : I suggest to rephrase; (pH 7.4) and finally diluted with water, in order to become an isotonic solution, ready for injection, (I did not understood, reversal osmose, here...)
line 110 : inoculated (instead of implanted),
line 128 : ...was calculated from caliper measurements...,
line 146 : ...at 10 min... - this should be 1 hour !!!, (to my opinion),
line 183 : performed (instead examined),
Author Response
Reviewer 2
(i) line 152-153 : statement: 'the pancreas expresses LAT1', is this really so ?, can this be underpinned by some references , [some references state, increased uptake in pancreas, for F18 labelled AA (Laverman et al., Eur J Nucl Med 2002, 29, 681 and Hanoaka, J Nucl Med...) but not always found ,everywhere ! ( i.e. *I-IPP, Baum et al., Nucl Med Mol Imaging 2011, (45), 299-307) ] - this should also mean, using *At-AAMP for cancer therapy, the pancreas will be exposed to a high radiation burden ! (or maybe useful for treatment of pancreatic adenocarcinoma, as there are some references in literature, with radio-labelled modified amino acids),
Reply:
LAT1 expression of murine pancreas was reported. We added the reference (ref. 18). In contrast, the human pancreas lacks LAT1 expression, thus high pancreatic accumulation of 2-211At-AAMP would not be expected in patients. Indeed, 18F-FAMT, a LAT1 specific amino acid tracer, showed high accumulation in the murine pancreas, but it accumulated less in human pancreas in clinical practice. We added this information and references in the Discussion section.
(ii) can probenecid really be used for clinical studies ?, can the translation be really made ? (as dose of probenecid, for mice is now 400 mg/kg), what is the opinion of the authors ?
Reply:
We understand your concern. 400 mg/kg is a high dosage in human terms. However, a smaller dose is expected to be effective in case of humans. It is well known that a normal dose of probenecid (2000 mg/day) can prolong plasma half-life and increase serum concentration of penicillin derivatives in humans. In addition, probenecid has similar effects on many other medications to increase serum concentrations of drugs such as acetaminophen, naproxen, indomethacin, ketoprofen, lorazepam and rifampin in patients. Thus, we think that normal dose of probenecid loading will improve the therapeutic effect of 2-211At-AAMP by enhancing its accumulation in tumors. We added the description about probenecid dose in the Discussion section.
minor :
Reply: Thank you for all your suggestions. We apologize for the many errors.
line 22 : is useful as adjuvant...,
Reply: Thank you for the suggestion. But in this sentence, we want to claim the usefulness of 2-211At-AAMP with probenecid loading, not usefulness of probenecid loading alone. Thus, we think it’s better not to change the sentence.
line 43 : delete 'anti-cancer' ,
Reply: We deleted 'anti-cancer'.
line 45 : promising (instead of great),
Reply: We changed it.
line 63 : ... an attractive radiohalogen α-emitter...,
Reply: We added ‘radiohalogen’.
line 76 : the clearance of radioiodine-labeled...,
Reply: We added ‘radio’ before ‘iodine’
line 90-91 : I suggest to rephrase; (pH 7.4) and finally diluted with water, in order to become an isotonic solution, ready for injection, (I did not understood, reversal osmose, here...)
Reply: Thank you for the suggestion. We revised the sentence according you suggestion. ‘reverse osmose’ is just mean using ‘reverse osmose water’ for dilution.
line 110 : inoculated (instead of implanted),
Reply: We changed it.
line 128 : ...was calculated from caliper measurements...,
Reply: We added the sentence.
line 146 : ...at 10 min... - this should be 1 hour !!!, (to my opinion),
Reply: We apologize for the confusing writing. This sentence means that the renal accumulation level in the probenecid preloading group was significantly lower at 10 min and significantly higher at 1 h and 3 h compared with in the control group. We changed ‘decreased’ to ‘lower’ and ‘increased’ to ‘higher’.
line 183 : performed (instead examined),
Reply: We changed it.
Reviewer 3 Report
The authors provide a study on the improvement of tumor targeting capability of a TAT agent. Targeted alpha therapy is a very very important topic and very complicated to be used. The paper is very well written in a clear and concise manner. The authors present clear objectives and a great discussion. However, I have a few questions/concerns mainly on the specificity of the radiotracer (and its tumor accumulation).
1) A significantly higher radiotracer accumulation was found in thyroid in the probenecid preloading group when compared to control group. The authors indicate that the uptake is not specific since it returns to "normal" after 6 hours. However, an uptake can still be specific and transient. Please discuss this a little more (cross reaction? the tracer, after binding to LAT1 would be internalized?)
2) On that note, would all the organs differential uptake (especially muscle and tumor) also be due to mere higher radiotracer amount in the blood (delayed clearance)?. because in fact, at later time point (6h), tumor and muscle uptake were not statistically significant when compared to controls. Could tumor uptake be higher only due to delayed clearance. This is very important. A blocking study or biodistribution studies with an isotype or non-targeted aminoacid analog should be used for comparison.
3) Is it possible to perform any additional ex vivo studies to correlate this uptake in the tumor? Immunohistochemistry (LAT1 expression) or conjugating AMP with a dye and verifying its presence/amount in the tumor tissue?
4) Please provide radiotracer uptake in other organs as well (heart, liver, spleen, bone, thyroid, stomach, intestine, pancreas, lung...) for the biodistribution in tumor bearing mice. Did the uptake in off-targeted tissue also significantly increased (liver? bone?)? Please also discuss.
5) Please provide individual tumor growth curves for the animals.
6) Since the blood circulation was longer with probenecid preloading, some toxicity concerns may arise. Other toxicity parameters should be carried out (or at least discussed) such as comprehensive blood chemistry metabolic panel, complete blood counts, IHC of major organs, etc. Radiotoxicity in the kidneys would be a major concern. Specially since probenecid might also be nephrotoxic (the weight loss in the probenecid group is due to what process? please discuss).
Author Response
Reviewer 3 1) A significantly higher radiotracer accumulation was found in thyroid in the probenecid preloading group when compared to control group. The authors indicate that the uptake is not specific since it returns to "normal" after 6 hours. However, an uptake can still be specific and transient. Please discuss this a little more (cross reaction? the tracer, after binding to LAT1 would be internalized?) Reply: Thank you for the comment. As the reviewer pointed out, accumulation in the thyroid might be specific one. We wanted say “it was not the specific accumulation of free 211At since free 211At is known to be retained in the thyroid”. Since thyroid accumulation level increased 3 h after injection (not early time point), this specific accumulation would not be LAT1-mediated. It is difficult to explain the results. We added the description and reference (ref 19) about thyroid accumulation of free 211At. 2) On that note, would all the organs differential uptake (especially muscle and tumor) also be due to mere higher radiotracer amount in the blood (delayed clearance)?. because in fact, at later time point (6h), tumor and muscle uptake were not statistically significant when compared to controls. Could tumor uptake be higher only due to delayed clearance. This is very important. A blocking study or biodistribution studies with an isotype or non-targeted amino acid analog should be used for comparison. Reply: Thank you for the valuable comment. We think the increasing uptake of 2-AAMP in all organs with probenecid would be due to higher radioactivity level in the blood. The reason of no significant difference at 6 h after injection is that the effect of probenecid was temporary and 2-AAMP was rapidly cleared after the effect wore off. In the control group, the accumulation level in all organs was already low at 3 h after injection (almost similar level compared to 6 h), while accumulation level in the probenecid loading group did not drop to that level at 3 h after injection. Thus, we consider that most of 2-AAMP was eliminated from the body by 3 h after injection in control group, while a certain amount of 2-AAMP was still remained at 3 h after injection and was eliminated by 6 h after injection in probenecid loading group, consequently the accumulation levels of the two groups became similar. As the reviewer suggested, additional biodistribution study would be useful to discuss the effect of probenecid. However, it is difficult to do blocking study since LAT1 takes up amino acids by exchange mechanism. Indeed, biodistribution of 2-[18F]fluoro-alpha-methylphenylalanine (2-18F-FAMP) was not changed by addition of 500 μg of nonradioactive 2-FAMP (J Label Compd Radiopharm. 2020;63:368–375). And it would be difficult to find a non-targeted amino acid analog which doesn’t accumulate in the tumor and is affected by probenecid. But we took this comment as important, we performed additional experiment. We checked LAT1 expression level with probenecid loading by Western blotting to confirm that probenecid has no effect on tumor cell. Probenecid treatment for 6 hours did not change the expression level of LAT1 in SKOV3 cells and other tumor cell lines. In addition, we reported that tumor and organs accumulation level of six different 18F-FAMPs highly dependent on blood radioactivity level (ref 11). Based on these results, an increase in tumor accumulation level of 2-AAMP by probenecid preloading is not caused by a direct effect on tumor cells but reflects delayed blood clearance. These results also indicate that the 2-211At-AAMP with probenecid loading method could be applied to other LAT1-positive cancers as well. We added the description and results for Western blotting in the Materials and Methods section and Results section. And we revised Discussion section as appropriate. 3) Is it possible to perform any additional ex vivo studies to correlate this uptake in the tumor? Immunohistochemistry (LAT1 expression) or conjugating AMP with a dye and verifying its presence/amount in the tumor tissue? Reply: Thank you for the suggestion. We checked LAT1 expression of the implanted tumor by Western blotting. Although we didn’t compare LAT1 expression level between with and without probenecid loading, it will be similar considering the results of in vitro Western blotting study. 4) Please provide radiotracer uptake in other organs as well (heart, liver, spleen, bone, thyroid, stomach, intestine, pancreas, lung...) for the biodistribution in tumor bearing mice. Did the uptake in off-targeted tissue also significantly increased (liver? bone?)? Please also discuss. Reply: According to reviewer’s suggestion, we added the data and explanation. Similar to the biodistribution study in normal mice, the uptake of 2-AAMP in other organs also significantly increased. We didn’t check all organs but accumulation of other organs would also significantly increase. The accumulation of 2-211At-AAMP for more organs has been reported previously (ref 13). 5) Please provide individual tumor growth curves for the animals. Reply: According to reviewer’s suggestion, we added the data. 6) Since the blood circulation was longer with probenecid preloading, some toxicity concerns may arise. Other toxicity parameters should be carried out (or at least discussed) such as comprehensive blood chemistry metabolic panel, complete blood counts, IHC of major organs, etc. Radiotoxicity in the kidneys would be a major concern. Specially since probenecid might also be nephrotoxic (the weight loss in the probenecid group is due to what process? please discuss). Reply: Thank you for the valuable suggestion. However, since limitation of 211At, it was difficult to do such kind of study. Base on the biodistribution study in normal mice, radiation-effective doses for humans were calculated. Compared to other study using radiopharmaceuticals with 211At, compare MABG data, current dose of 2-211At-AAMP would not cause severe toxicity. As the reviewer pointed out, probenecid would cause body weight loss. But body weight loss or nephrotoxicity isn’t reported as an occasional side effect in clinical practice. Thus, it is difficult to explain the reason of the weight loss. But since probenecid will be used at normal dose in clinical practice, it is not expected to cause severe toxicity. We added the description about dosimetry and discuss about toxicity in the Discussion section.